# Bud-Poplar-Extract-Embedded Chitosan Films as Multifunctional Wound Healing Dressing

**DOI:** 10.3390/molecules27227757

**Published:** 2022-11-10

**Authors:** Carla Russo, Miranda Piccioni, Maria Laura Lorenzini, Chiara Catalano, Valeria Ambrogi, Rita Pagiotti, Donatella Pietrella

**Affiliations:** 1Medical Microbiology Unit, Department of Medicine and Surgery, University of Perugia, Piazzale Sereni, Building D, 4th Floor, 06129 Perugia, Italy; 2Biochemical Sciences and Health Unit, Department of Pharmaceutical Sciences, University of Perugia, Via del Giochetto, 06122 Perugia, Italy; 3Pharmaceutical Technology Unit, Department of Pharmaceutical Sciences, University of Perugia, Via del Liceo 1, 06123 Perugia, Italy

**Keywords:** bud poplar resin extract, chitosan, wound healing, antibiofilm activity, antioxidant activity, anti-inflammatory activity

## Abstract

Wounds represent a major global health challenge. Acute and chronic wounds are sensitive to bacterial infection. The wound environment facilitates the development of microbial biofilms, delays healing, and promotes chronic inflammation processes. The aim of the present work is the development of chitosan films embedded with bud poplar extract (BPE) to be used as wound dressing for avoiding biofilm formation and healing delay. Chitosan is a polymer with antimicrobial and hydrating properties used in wound dressing, while BPE has antibacterial, antioxidative, and anti-inflammatory properties. Chitosan-BPE films showed good antimicrobial and antibiofilm properties against Gram-positive bacteria and the yeast *Candida albicans*. BPE extract induced an immunomodulatory effect on human macrophages, increasing CD36 expression and TGFβ production during M1/M2 polarization, as observed by means of cytofluorimetric analysis and ELISA assay. Significant antioxidant activity was revealed in a cell-free test and in a human neutrophil assay. Moreover, the chitosan-BPE films induced a good regenerative effect in human fibroblasts by in vitro cell migration assay. Our results suggest that chitosan-BPE films could be considered a valid plant-based antimicrobial material for advanced dressings focused on the acceleration of wound repair.

## 1. Introduction

Skin is the largest organ of the body. It protects from external insults such as ultraviolet radiation, toxic chemical compounds, and microorganisms. A wound is defined as the interruption of the anatomic skin tissue; it may be classified as acute or chronic depending on the healing time. While an acute wound may heal completely within 8–12 weeks, chronic wounds usually take longer periods (more than 12 weeks) with a high possibility of recurrence. Chronic wounds represent a major global health challenge. Diabetic foot ulcers are among the most common complications of diabetes mellitus, with a lifetime incidence of up to 15% among the diabetic population [1]. In 2017, 425 million people worldwide, or 8.8% of adults aged 20–79 years, were estimated to have diabetes. If the current trend continues, by 2045, 629 million people aged 20–79 years will have diabetes. Venous insufficiency and peripheral neuropathy lead to loss of sensitivity in the feet, which degenerates into chronically infected lesions known as diabetic foot ulcers. Moreover, pressure ulcers are a serious problem for all bed-bound and chair-bound patients. Wound infections may also occur in burn victims, patients with traumatic wounds, and patients with surgical wounds. Both acute and chronic wounds are sensitive to bacterial infection, which can also elicit a systemic response. This leads to the retardation or inhibition of the healing process. The wound environment facilitates the development of microbial biofilms. Biofilms are found on the surface of the skin, and there is much evidence to suggest their involvement in the delay of wound healing and the chronic inflammation process [2]. The prevention of biofilm formation is the goal of wound treatment because the standard protocols based on topical and systemic administration are often unable to remove biofilms [3].

Wound dressings are designed to protect the wound from the external environment and to absorb wound exudate during the healing process. Many types of materials have been utilized to develop wound dressings and been commercialized in the market. Bio-derived polymers such as polysaccharides have been tested in wound dressing development; they are biocompatible, nonimmunogenic, and have anti-microbial properties [4].

Chitosan is a biocompatible and biodegradable polymer with antimicrobial and hydrating properties [5], obtained from the partial deacetylation of the natural polymer chitin. Chitosan has been approved for wound dressing applications. Chitosan nanoparticles show a positive surface charge and mucoadhesive properties that allow their adherence to mucus membranes and the release of the drug payload in a sustained release manner [6]. Chitosan-based dressings offer several advantages over traditional materials employed in the wound healing process [7]. Chitosan-based dressing materials are considered suitable for clinically problematic wounds, as they facilitate hemostasis, enhance wound healing, and have antimicrobial properties. Moreover, chitosan showed low toxicity and immune-stimulatory activity [8]. Chitosan displays excellent film-forming capability and finds many applications in oral mucosal, buccal, transdermal, sublingual, and periodontal delivery systems as a carrier for drugs and bioactive agents ranging from small molecule-like antibiotics, nucleic acid, and proteins [9]. The incorporation of bioactive components in wound dressings could improve wound dressing performance [10]. Propolis is one of the biocidal molecules used with chitosan. Moreover, chitosan is similar to glycosaminoglycans with similar morphological and mechanical properties to collagen; these features make it an excellent alternative for tissue engineering applications.

Bees use propolis to keep a low concentration of bacteria and fungi in the hive. Thus, the action against microorganisms is an essential characteristic of propolis exploited by man since ancient times. It is still one of the remedies in the Balkan states for the treatment of wounds and burns [11]. Propolis is widely used in traditional medicine for its antiseptic, antibacterial, antioxidative, anti-inflammatory, and antitumor properties [12,13,14,15].

A twenty percent extract of Malaysian propolis in chitosan nanoparticles showed an antibiofilm effect against *Enterococcus faecalis*. The chitosan–propolis nanoformulation downregulated the expression of cytolysin and biofilm-forming genes, rendering the bacteria susceptible to treatment, suggesting that the nano formulation is an ideal antibiofilm agent [16]. Moreover, propolis promotes skin wound healing by stimulating epithelial re-generation, modulating extracellular matrix deposition, and facilitating the formation of granulation tissue [17].

The antimicrobial components of propolis are different in distinct geographic regions. For example, in European samples, flavonoids and cinnamic acid derivatives have been observed, and in Brazilian propolis, diterpenic acids and prenylated coumaric acids have been identified [11]. The composition of propolis is chemically related to that of the bud exudates from which it derives. In temperate climates, bees are thought to collect propolis mainly from Populus but also from *Betula* (birch), *Salix* (willow), *Alnus* (alder), and *Aesculus* (horse chestnut) [18]. Bud exudates of *Populus* species (cottonwood, poplar, aspen) are the main propolis plant sources. Poplar buds are coated with a viscous substance, an exudate, which has been reported to contain phenolic compounds (terpenoids, flavonoid aglycons and their chalcones) and phenolic acids and their esters [19]. Poplar buds have been used as a remedy in traditional medicine. Leaf buds were used as an expectorant and in the treatment of dermatitis, while leaves and bark have demonstrated antirheumatic properties. The leaf buds of *Populus nigra*, *P. canadensis*, and *P. balsamifera* have shown antioxidant properties [20]. Moreover, Wang et al. showed that *P. canadensis* bud extracts have significant anti-inflammatory effects by inhibiting IL-6, IL-10, MCP-1, and TNF-α secretion and blocking NF-κB activation [21].

This work aims to develop eco-friendly films based on chitosan-containing bud poplar resin extracts to be used as a wound dressing.

## 2. Results

In our recently published work, we have characterized and analyzed the extract of poplar buds that showed antioxidant, anti-inflammatory, and antibiofilm activity against the Gram-negative bacterium Pseudomonas aeruginosa [13]. In the present work, the same extract was incorporated into chitosan films. The films were first characterized, and then their antimicrobial, antibiofilm, antioxidant, anti-inflammatory, and regenerative properties were determined.

### 2.1. Characterization of Chitosan–Bud Poplar Resin (CBPR) Films

The photographic images of the prepared films are reported in Figure 1. As can be observed, chitosan film was transparent and clearer than the other films, which were gradually darker at the higher end of the bud poplar extract concentration. A good dispersion of the extract was obtained up to 1% concentration, and at 2%, aggregates of the extract were observed. Chitosan film thickness was 60 ± 10 μm, and it was not affected by the addition of the bud poplar extract.

The ATR FT-IR spectra of the CBPR 2%, chosen as a model film, the plain chitosan film, and the bud poplar extract are reported in Figure 2. Chitosan film showed the typical polymer absorptions bands. The large band at 3270 cm^−1^ is associated with -OH and NH_2_ stretching vibrations, the absorption peaks at 1634 cm^−1^ and 1554 cm^−1^ are assigned to the NHCO (amide I) and -NH bending (amide II), respectively, and finally the absorption bands at 2922 cm^−1^ and 2869 cm^−1^ are relative to CH_2_ bending [22]. The bud poplar extract spectrum shows the typical hydrogen-bonded OH stretch of phenolic compounds at 3336 cm^−1^. Other bands can be observed at 2917 cm^−1^ and 2849 cm^−1^ due to the asymmetric and symmetric stretching of CH_2,_ at 1157 cm^−1^, attributable to C-O and C-OH vibration, at 1448 cm^−1^ due to C-H vibration, 1512 cm^−1^, 1604 cm^−1^, and 1630 cm^−1^, attributed to aromatic ring deformations, and at 1681 cm^−1^ due to C=O stretching of flavonoids and lipids [23,24].

The chitosan and CBPR 2% films’ spectra had almost the same patterns, and no shifts of chitosan-typical absorption bands could be observed, indicating the absence of interactions between propolis and chitosan. In CBPR 2%, spectrum absorption bands of propolis could not be observed, possibly due to the low concentration of bud poplar extract in the film.

In Figure 3, the surface appearance of neat chitosan film and CBPR 2% film are reported. As shown, the surface of the chitosan film was quite homogeneous, and after incorporation of bud poplar extract, no change in microstructure could be observed.

CBPR films were characterized for water absorption capability and bud poplar resin in vitro release. Water absorption capacity is an important property for a wound dressing material, as it absorbs wound exudates, avoiding maceration and enhancing wound healing.

All films showed a good and fast hydration capacity (Figure 4). After 30 min, hydration was in the range of 120–140%. The presence of bud poplar resin did not decrease the water sorption capability; rather, the best hydration percentage values were obtained with CBPR 1% and CBPR 2% films.

Bud poplar resin was released very slowly, and after 24 h, the percentage of the release was 13%, 17%, and 37% for CBPR 2%, CBPR 1%, and CBPR 0.5% films, respectively (Figure 5A), corresponding to 8 µg/mL, 5.5 µg/mL, and 5 µg/mL (Figure 5B). After 24 h, the release showed only a very small increase, though no plateau was reached. The initial faster release of bud poplar resin can be explained by the release of the BPE extract fraction present on the surface of the films, which was released faster than that present deeper in the films.

### 2.2. Antimicrobial Activity of CBPR Films

After CBPR films were characterized, their antimicrobial activity was determined with a Kirby–Bauer radial diffusion assay (Table 1). Chitosan films induce a contact inhibition halo for all microorganisms, while an evident inhibition zone is recorded in all microorganisms treated with the CBPR 2% films. The inhibition zone increase was dose-dependent with respect to BPE in the film. These data are very interesting because they suggested that in the areas in direct contact with the film, there was no microbial proliferation, and in the presence of the poplar bud extract the antimicrobial activity could reach deeper layers.

The impact of biofilms on chronic wound infection and their involvement in late healing is well established [25]. Starting from this assumption, the ability of the CBPR films to slow down the formation of the biofilm and their ability to disperse already-formed microbial biofilms were evaluated. All films were active during the formation of *S. aureus, S. epidermidis*, and *C. albicans* biofilms (Figure 6); the antibiofilm activity was always significant with respect to the untreated sample. Chitosan films were able to inhibit the biofilm formation of staphylococci, while the addition of bud poplar resin extracts at 0.5%, 1%, and 2% did not affect this activity, although a tendency to reduce the biofilm of *S. aureus* was evident. No effect was observed on biofilm formation or dispersion of Gram-negative bacterium *Pseudomonas aeruginosa*. Interestingly, chitosan films were able to disperse preformed *S. epidermidis* and *C. albicans* biofilms, while films containing chitosan and BPE were able to increase the dispersion of all preformed biofilms of *S. aureus* and *Candida* yeasts. The CBPR 1% and 2% dispersed the preformed biofilm of *S. aureus* better than chitosan film. The CBPR 2% film was able to significantly increase the dispersion of the *Candida* biofilm compared to the biofilm treated with chitosan film.

### 2.3. Regenerative Activity of CPBR Films

Wound healing is a complex and dynamic process. Ulcers are caused by a prolonged inflammatory process, persistent infections produced by microbial biofilms, and failure of the epidermal and/or dermal cells to respond to the reparatory stimuli. Chitosan is approved as safe by the European Medicines Agency and the U.S. Food and Drug Administrations for wound dressing applications [6]. Cytotoxicity of CBPR films was tested on human dermal fibroblast (HuDe) and human keratinocyte (NCTC2544) cell lines. The results reported in Table 2 show that all tested CBPR films were not cytotoxic after four hours of contact with fibroblasts, while after 24 h, the 1% and 2% BPE decreased the viability of cells by 27% and 55%, respectively. Keratinocytes were more sensitive to chitosan films than were the fibroblasts; after 4 h of contact, CPBR 1% and 2% reduced the viability of cells by 45% and 64%, respectively, and after 24 h a reduction of 70% was observed. The CPBR 0.5% film was not cytotoxic for either of the two cell lines after 4 h and 24 h.

The regenerative capability of bud poplar extract (BPE) on human dermal fibroblasts (HuDe) was assessed by a mobility assay that used a silicone apparatus (Ibidi Micro Insert 4-well dishes) to grow the cells of the dermis at a predefined distance. The removal of the silicone support allowed quantifying the closure of spaces (Figure 7A) in the presence or absence of CBPR 0.5%. Cell distances were monitored by microscopy and measured for 7 days (Figure 7B,C). The results showed that BPE at low concentrations induces a regenerative activity by chitosan films on fibroblasts of the human dermis. The effect ©s already evident after 24 h, and it is maintained until the closure of the septum at 7 days.

### 2.4. Anti-Inflammatory Activity of BPE

Given the known immunomodulatory properties of propolis, we analyzed how the BPE that can be released by CBPR films can influence the polarization of macrophages towards an anti-inflammatory M2 phenotype. To this end, we tested a model of human macrophage cell line THP-1 stimulated with phorbol-12-myristate-13-acetate (PMA) [26]. PMA treatment, which activates protein kinase C (PKC), induces a greater degree of differentiation in THP-1 cells as reflected by increased adherence and expression of surface markers associated with macrophage differentiation. During this process, the cultured cells without PMA (PMAr) increased their cytoplasmic-to-nuclear ratio, mitochondrial and lysosomal numbers, and altered differentiation-dependent cell surface markers in a pattern similar to MDM (monocyte-derived macrophages). Moreover, PMAr cells showed relative resistance to apoptotic stimuli and maintained levels of the differentiation-dependent antiapoptotic protein Mcl-1, similar to MDM [26]. At the end of the treatment, we obtained an M0 phenotype. To evaluate the direct immunomodulatory effect of BPE, M0 cells were treated with BPE extract (5, 25, and 50 μg/mL). No cytotoxicity of BPE extract was observed on M0 and M1 cells (Appendix A). The polarization was evaluated by the cytofluorimetric determination of the membrane marker receptors of the two populations: B7 molecules (CD80 and CD86) for the inflammatory phenotype M1 and CD36 for the M2 anti-inflammatory phenotype. The results in Figure 8 show that the extract was not able to induce polarization of M0; even BPE at a concentration of 50 µg/mL was able to slightly reduce the expression of CD36. However, B7 molecule expression is not affected by the extract.

To evaluate the anti-inflammatory activity, M0 were treated for 24 h with BPE extract before or after LPS/IFNγ stimulation to induce the M1 phenotype. Pretreatment of M0 with BPE at 5–25–50 µg/mL before stimulation with LPS/IFNγ did not regulate the expression of the proinflammatory B7 molecules (Figure 9A), but instead significantly increased the expression of the anti-inflammatory M2 receptor CD36 (Figure 9C). When macrophages were first polarized towards M1 phenotype, BPE treatment did not significantly affect the expression of B7 or CD36 molecules (Figure 9B,D).

To further verify the immunomodulatory activity of the extract, we evaluated the secretion of proinflammatory and anti-inflammatory cytokines by macrophages stimulated with BPE before M1–M2 polarization. BPE at 5, 25, and 50 µg/mL were not able to induce the proinflammatory cytokine TNFα (Figure 10A). Regarding the production of the anti-inflammatory IL-10, we verified that poplar bud extract is not able to stimulate the production of the cytokine (Figure 10B), although an increase is observed with the highest dose used (50 µg/mL). Secondly, we analyzed the production of TGFβ1; the extract at a concentration of 50 μg/mL was able to stimulate the production of the cytokine by the M0 cells (Figure 10C), suggesting an M2 polarization. Ansorge et al. have described a similar pattern analyzing the effect of propolis and its constituents on human T regulatory cells and observed that TGFβ1 production was increased [27].

### 2.5. Antioxidant Activity of BPE and CBPR Films

Bud poplar extract showed strong antioxidant activity due to its abundant active poly-phenols [15,18]. Moreover, it is known that the antioxidant activity of propolis contributes to its protective effects on skin diseases; for example, it was reported that propolis could alleviate cell damage in fibroblasts by suppressing intracellular ROS production. Moreover, a lower concentration of free radicals has been detected in burn wounds treated with propolis [15]. In the first series of experiments, we analyzed the antioxidant ability of our BPE by means of neutralizing the free DPPH radical. BPE at a concentration of 500 μg/mL showed a slight scavenging ability versus the positive control ascorbic acid (Figure 11).

In a second series of experiments, we analyzed the antioxidant ability of chitosan film by means of neutralizing the free DPPH radical. Chitosan film itself showed antioxidant activity, while only CBPR 2% significantly increased the scavenging activity of chitosan to 70% (Figure 12).

Total ROS production by human neutrophils stimulated with phorbol-12-myristate-13-acetate (PMA) was tested by chemiluminescence assay. Neutrophils activated by PMA produce luminol-dependent chemiluminescence profiles following ROS production after the addition of the stimulus. The results showed that bud poplar resin extract was able to reduce ROS production by neutrophils activated with PMA in a dose-dependent manner (Figure 13). This result is very encouraging, as it demonstrates that the activity of DPPH scavenging of the extract is enhanced by its ability to lower ROS concentration in the supernatants of human PMN, cells recruited in the infection site during the inflammation process.

As observed with the DPPH test, chitosan film is able to reduce the production of ROS. However, CBPR films decreased the ROS concentration in BPE in a dose-dependent manner (Figure 14).

Then, the antioxidant activity of BPE on human fibroblasts and keratinocytes stimulated with IFNγ and histamine was analyzed. BPE, by themselves, did not induce the ROS production in human dermal fibroblasts (Figure 15A). However, BPE at the concentration of 5 µg/mL, but not at 50 µg/mL, reduced the ROS formation induced by IFNγ/histamine treatment. These results suggest a dual activity of the extract at different concentrations. It seems, in fact, that at lower doses, the anti-inflammatory activity predominates, while at higher doses the proapoptotic and antiproliferative activity described in the literature is evident [28].

Human keratinocytes (NCTC 2544) produce a high basal level of ROS; stimulation with BPE reduced in a dose-dependent manner the production of radicals (Figure 15B). Similar results have been reported by Kim et al. that showed the reduced generation of ROS in human keratinocytes pretreated with propolis [29]. The effect could be ascribed to chrysin as suggested by Wu et al., although it was not detected in our extract [30].

## 3. Discussion

Keeping the wound environment as sterile as possible helps to prevent the entrance of microorganisms into the wound and the seeding of biofilm infections. Wound disinfection does not eliminate microorganisms. Wound dressing is an important tool for accelerating healing and avoiding microbial colonization [31]. Many plant compounds with antioxidant, anti-inflammatory, and antimicrobial properties could be of great benefit for wound healing using bioactive wound dressings [32].

Our data about the antimicrobial activity of BPE extract agree with those obtained by Popova et al., who observed better activity of propolis towards Gram-positive bacteria compared to Gram-negative bacteria [11]. Moreover, Popova et al. [11] analyzed the antibacterial activity of propolis from different regions of Turkey, confirming the importance of the amount of phenolic compounds, flavones, and flavanones for the antibacterial activity of poplar propolis. The extract used in the film contains about 25% of the total flavonoid [13]. These results are in line with those observed by Torlak et al. [33] and Eskandarinia et al. [23]. The antibacterial efficacy of chitosan–propolis-coated polypropylene films has been studied against foodborne pathogens *Bacillus cereus*, *Cronobacter sakazakii*, *Escherichia coli* O157:H7, *Listeria monocytogenes*, *Salmonella typhimurium*, and *Staphylococcus aureus* [17,34,35,36]. Chitosan-coated film exhibited a broad spectrum of antibacterial activity; the incorporation of ethanol propolis extract enhanced antibacterial activity against all pathogens tested [33]. The recent work of Eskandarinia et al. described the antimicrobial activity of cornstarch, hyaluronic acid, and propolis wound dressing [23]. Film dressing containing propolis exhibited higher antibacterial activity against *S. aureus*, *E. coli,* and *S. epidermidis* in comparison with the cornstarch and cornstarch/hyaluronic acid dressings. Moreover, our data align with the results described in a recent study that demonstrated that chitosan–propolis nanoparticles were able to reduce the biofilm formation of *Staphylococcus epidermidis* [16].

Recently, propolis has been used as an active component in wound biomaterials for its antimicrobial effects, but to the best of our knowledge, the antibiofilm, anti-inflammatory, and cell proliferative activities of chitosan–BPE films have not been evaluated [23,24,37,38,39,40,41,42]. When formulated in corn-starch-based wound dressing with hyaluronic acid, the proliferation of mouse fibroblasts was observed, but this was due to the presence of hyaluronic acid and actually decreased in films containing propolis [30]. This latter result is in contrast with the regenerative activity of our BPE in chitosan film. However, we used human dermal fibroblasts, while in the previous study, murine fibroblasts from subcutaneous adipose tissue were used. Moreover, contrary to Paramasivan et al., who have described an antiproliferative activity of chitosan–dextran gel on human fibroblasts [43], the chitosan included in our film showed no activity on the proliferation of human fibroblasts, confirming that the active CBPR–chitosan film could be an excellent dressing for wounds.

Regarding the anti-inflammatory activity of BPE, Bueno-Silva et al. reported the effect of propolis on the expression of B7 molecules; they stimulated murine macrophages with Brazilian red propolis extract and demonstrated an up-regulation of the costimulatory molecules [14]. The apparent discrepancy with our data is due to the different composition of the propolis compared to bud poplar extract and the cells used; indeed, the authors have tested the propolis extracts on murine macrophages, while in our study, the effect of the extract on human macrophages was analyzed. To date, as far as we know, this is the first work studying the activity of poplar bud extract on the polarization of human M1/M2 macrophages, confirmed by the production of the anti-inflammatory cytokine TGFβ1, the slight increase of IL-10, and decrease of the proinflammatory cytokine TNFα.

In a recent work on propolis originating in different countries, Conti et al. [44] demonstrated that Brazilian and Cuban propolis increase TNFα production by human monocytes, while Mexican propolis decreases the basal production of the cytokine. The authors attributed this difference to the different composition of the propolis; in particular, the major compounds found in Brazilian, Cuban, and Mexican propolis samples were artepillin C, isoflavonoids, and pinocembrin, respectively.

Macrophages exposed to LPS are characterized by an IL-10-high and IL-12-low phenotype and promote type II responses; they have been called M2b [45]. In presence of IL-10, macrophages polarize towards the M2c phenotype that is involved in immunoregulation, matrix deposition, and tissue remodeling.

The antioxidant activity of propolis has been widely studied; propolis varieties of different geographic locations have shown an antioxidant activity between 20% and 67% at the concentration of 20 µg/mL, showing that they are more active than our extract [46]. Kurek-Górecka et al. [47] reported antioxidant activity of propolis at concentrations much higher than that used in our experiments. Finally, Bonamigo et al. [48] reported antioxidant activity of propolis extracts with IC50 values between 50 and 180 µg/mL.

The addition of BPE to chitosan films improved the antioxidant activity in a dose-dependent way. The flavonoid content (pinocembrin and pinostrobin) of poplar bud extracts has been shown to be responsible for its antioxidant effect [19,20]. Pinocembrin is present in our extract to a greater extent than in propolis [13], thus confirming its excellent antioxidant activity.

## 4. Materials and Methods

### 4.1. Preparation of Chitosan–Bud Poplar Resin (CBPR) Films

Chitosan film was prepared by dispersing 1.5% (*w*/*w*) chitosan in a solution of 0.5% (*v*/*v*) acetic acid, 0.1% (*v*/*v*) glycerin, and 1% (*w*/*w*) ethanol 96°. Bud poplar resin extracts (BPE) in 85% ethanol were graciously provided by ABOCA S.p.A. (Arezzo, Italy). Poplar buds were grounded by an electronic blender before extraction. Samples were then centrifuged; supernatants were recovered, concentrated, and lyophilized. Samples were maintained at 4 °C. A stock solution of each bud poplar extract was prepared in ethanol at a concentration of 100 mg/mL and maintained at 4 °C. The extract was characterized as previously described [13].

Films containing bud poplar resin were prepared by mixing 1.5% chitosan with an aqueous solution of 0.5% (*v*/*v*) acetic acid, 0.1% (*v*/*v*) glycerin, and 1% (*w*/*w*) ethanol 96°, in which an appropriate amount of bud poplar resin extract had been previously dissolved. Films containing 0.5%, 1%, and 2% w/w BPE were prepared. These are henceforth referred to CBPR 0.5%, CBPR 1%, CBPR 2%.

The mixture was stirred for 24 h until a homogeneous dispersion was obtained. Subsequently, the air in the dispersion was removed by using a vacuum pump, and the mixture (20 g) was seeded in Petri dishes (diameter of 10 cm) and left to rest for 24 h. Films were then dried at 50 °C for 6 h.

### 4.2. ATR-FT-IR

ATR-FT-IR spectra were recorded in transmittance using a Spectrophotometer Shimadz QATR-S IR Spirit A552. The wavenumber range was from 4.000 to 400 cm^–1^ with a resolution of 4 cm^−1^. All spectra were rationed against the spectra of an empty cell.

### 4.3. Field Emission Scanning Electron Microscopy (FE-SEM)

FE-SEM examined the morphology of the films through a LEO 1525 ZEISS instrument (Oberkochen, Germany). The acceleration potential voltage was maintained at 1 keV. Samples were placed onto carbon-tape-coated aluminum stubs. The stubs were sputter-coated with chromium before imaging by a high-resolution sputter (Quorum Technologies, East Essex, UK). The coating was performed at 20 mA for 20 s.

### 4.4. Water Absorption of Films

Film squares of 2 × 2 cm were stabilized in a dryer in the presence of a saturated solution of magnesium nitrate (RH 53%). Before the assay, the films were sterilized and weighed (W1), then films were immersed in Petri dishes containing a simulated wound fluid (SWF 0.02 M calcium chloride, 0.4 M sodium chloride, 0.08 M tris hydroxymethyl, aminomethane in deionized water, pH 7.5) [49,50]. The capsules were then incubated at 37 °C (±0.1). At fixed time points, the films were recovered from the fluid, and after absorption of excess liquid, they were weighed again (W2). Hydration was determined by calculating the change in weight using the following formula:% Hydration. = [(W2t − W1)/W1] × 100
W1 = weight of the initial dried film
W2t = weight of the wet film at different time points (t)

Tests were made in triplicate, and results are reported as an average ± SD.

### 4.5. Release Test

Bud poplar resin release studies were performed by immersing circular films of CBPR 2% (2 cm diameter) in 10 mL of a mixture composed of SWF/ethanol 80/20 *v*/*v*. The presence of ethanol guaranteed sink conditions. The test was performed at 37 °C under mild stirring (80 rpm). At fixed time intervals, aliquots (1 mL) of supernatant were removed and replaced by a fresh dissolution medium. Bud poplar resin concentration was determined by UV spectrophotometry using an Agilent 8453 UV–Vis spectrophotometer at a maximum absorbance of λ_max_ = 290 nm [37]. The experiment was done in triplicate, and results are reported as an average.

### 4.6. Microorganisms

The microbial strains used in this study were the two Gram-positive bacteria *Staphylococcus aureus* (ATCC 29213) and *Staphylococcus epidermidis* (ATCC 35984), the Gram-negative *Pseudomonas aeruginosa* (ATCC 15692), and the yeast *Candida albicans* (SC5314). The bacterial cultures were maintained in Muller–Hinton agar (MHA). The day before the test, one colony was inoculated in Mueller–Hinton broth (MHB) and incubated for 24 h at 37 °C. *Candida* cells from stock cultures in Sabouraud dextrose agar (SDA) with 50 μg/mL chloramphenicol were grown in Sabouraud broth at 37° C for 24 h. Microbial cells were harvested by centrifugation, washed, counted by spectrophotometric analysis (600 nm), and brought to the desired concentration in the appropriate culture medium.

### 4.7. Kirby–Bauer Assay

The antimicrobial activity of CBPR films was measured by Kirby–Bauer assay in MHA (Gram-positive *S. aureus* and *S. epidermidis*, Gram-negative *P. aeruginosa*) or in SDA Agar (*Candida*) Petri dishes.

Microbial suspensions were prepared from an 18 h culture in a liquid medium. The microorganisms were recovered in sterile phosphate buffer (PBS) and brought to a concentration of 10^8^ CFU/mL.

Dishes were seeded with the different microbial strains by sliding (rotating the plate 60° three times). The 6 mm diameter UV-sterilized film disks were hydrated with 20 μL of sterile water to facilitate both the placement on the surface of Petri dishes seeded with the microorganisms and to promote the release of the active ingredients from the film. A film of only chitosan was used as a negative control. Filter paper disks containing gentamicin (30 μg, Oxoid) for bacterial strains and fluconazole (25 μg, Oxoid) for *C. albicans* were used as positive controls. After 24 h of incubation at 37 °C, the microbial growth inhibition halos were measured. The extent of inhibition (80% growth reduction) was expressed as the diameter of the halo zone in mm.

### 4.8. Biofilm Determination

The ability of CBPR films to inhibit the growth of bacterial biofilms of *S. aureus*, *S. epidermidis*, *P. aeruginosa*, and *C. albicans* was evaluated as previously described [51] with some modifications. Bacterial cells were grown on MHB broth medium for bacteria or SAB broth for yeast overnight at 37 °C. The following day, microorganisms were diluted in 2% sucrose MHB or 2% sucrose SAB at the concentration of 10^6^–10^7^ cells/mL and inoculated (100 μL) in a 96-well polystyrene flat-bottom plate. Chitosan-based films 6 mm in diameter, previously sterilized with UV rays, were layered on microbial suspensions, and finally, 100 μL of medium were added to each well. Gentamicin (20 μg/mL) and fluconazole (20 μg/mL) were added as a positive control for bacterial strains and *C. albicans*, respectively. Plates were incubated for 24 h at 37° C. For dispersion study, biofilms were grown for 24 h in a 96-well plate, then CBPR films were added and incubated for a further 24 h at 37 °C. After removal of the biofilm, it was quantified by crystal violet staining [51]. The absorbance was measured at 570 nm. All samples were tested in quadruplicate, and each experiment was repeated at least twice. The data were expressed as a mean ± SD of the percentage of the biofilm mass calculated with respect to untreated biofilm.

### 4.9. Cytotoxicity Assay

After treatment with CBPR films, the viability of HuDe (human dermis fibroblast cell line, Istituto Zooprofilattico Sperimentale della Lombardia e dell’Emilia Romagna, BS PRC 41) and NCTC 2544 (human skin keratinocytes, Istituto Nazionale per la Ricerca sul Cancro HL97002), maintained in RPMI 1640 medium plus 10% FCS and penicillin 100 IU/mL and streptomycin 100 μg/mL (cRPMI), was measured using the ViaLight Plus Kit (Lonza, ME, USA). Cytotoxicity effect of BPE (5, 25, and 50 µg/mL) was determined on THP-1 cells (human monocyte-macrophages) in cRPMI, and the percentage of live cells was determined by ViaLight Plus Kit.

### 4.10. Human Macrophages Polarization M1/M2

The THP-1 cell line was maintained in cRPMI in a humidified incubator containing 5% CO_2_ at 37 °C. THP-1 cells were seeded at the concentration of 2 × 10^5^/tube in 1 mL of cRPMI and were differentiated into macrophages (M0) using 50 ng/mL (80 nM) of PMA for 48 h. The PMA-containing media was aspirated gently from the activated M0, and cells were incubated in fresh cRPMI for a further 5 days. To study the immune-modulating effect of poplar bud extracts, M0 were treated with BPE (5, 25, 50 µg/mL) for 24 h. After incubation, cell suspensions were centrifuged, supernatants were recovered and stored at −20 °C for cytokine determination, and pellets were suspended in PBS and transferred to a 96-well plate where cells were fixed with paraformaldehyde (PFA) 2% (100 µL for each well) for 15 min at room temperature. After fixation, the plate was centrifuged, and cells were labelled at 4 °C for 30 min with specific antibody mouse antihuman CD36 for M1 phenotype and mouse antihuman CD80 and CD86 for M2 phenotype in 100 μL of fluorescent buffer (PBS + 2% FBS + 0.1% sodium azide). After labelling, cells were re-suspended in 100 μL of focusing fluid and transferred in tubes containing other 900 μL of focusing fluid. All samples were read at the flow cytometer (Attune NxT, Life Technology, Thermo Fisher Scientific, Milan, Italy).

### 4.11. Cytokine Determination

Culture supernatants were recovered and stored at −20 °C until cytokine determination. TNFα, TGFβ1, and IL-10 were determined using an immune assay Enzyme-Linked ImmunoSorbent Assay (ELISA) (U-Cytech Biosciences, Utrecht, The Netherlands). The ELISA test was performed in three days. On the first day, a 96-well ELISA plate was prepared by adding a specific coating antibody and then was incubated overnight at 4 °C. The day after, the coating antibody was removed, and the plate was washed 4 times with wash buffer (0.05% Tween-20 in PBS). Blocking buffer (10% bovine serum albumin in PBS) was added to the plate, then the plate was incubated for 1 h at 37 °C. The blocking buffer was removed, and specific standards and samples were added to the wells. The plate was incubated at 4 °C overnight. On the last day, the standards and samples were removed, and the plate was washed 4 times with wash buffer. Specific detection antibody was added to the wells, and the plate was incubated for 1 h at 37 °C. The detection antibody was removed, the plate was washed 4 times, and the SSP conjugate antibody was added to the wells. After 1 h of incubation at 37 °C, the plate was again washed 4 times, and 3, 3′, 5, 5′-tetramethylbenzidine solution was added to the wells. After 20 min in darkness at room temperature, the solution turned blue; the stop solution was added, and the color became yellow. The plate was then read after 30 min at 450 nm (Infinite M200pro, Tecan, Milan, Italy).

### 4.12. Human Polymorphonuclear (PMN) Cells Isolation

Heparinized venous blood was obtained from a buffy coat gently provided by the Blood Bank of the Ospedale della Misericordia of Perugia. All donors have been informed and they signed the consensus form (MO-SIT_06) approved by Ethics Committee CEAS (Comitato Etico Aziende Sanitarie) (Rev. 3 Ottobre 2014) in which they authorize the use of their sample for research studies. Heparinized venous blood was diluted with RPMI 1640 (Gibco-BRL). Human PMN were recovered by density gradient centrifugation over Ficoll-Hypaque Plus (Amersham Pharmacia Biotech, Milan, Italy), counted, and adjusted to the desired concentration.

### 4.13. Evaluation of ROS Production by Chemiluminescence Assay

Antioxidant activity was evaluated by chemiluminescence assay according to the previous report [13] with small modifications. PMN were then stimulated with 2 × 10^−8 ^ M PMA in the absence or presence of CBPR films. The chemiluminescence produced by the cells was monitored for 20 min in a luminometer (Infinite M100 Tecan). The light output was recorded as RLU (relative photons light units). Each film was tested in triplicate in two separate experiments.

### 4.14. DCF Method for Detection of Intracellular ROS

The normal human keratinocyte cell line NCTC 2544 and the human dermis fibroblast cell line HuDe were routinely maintained in cRPMI and incubated at 37 °C in a humidified incubator containing 5% CO_2_. The medium was changed every 2–3 days. For experiments, the cells were trypsinized and plated either in 96-well plates (2 × 10^5^ cells/mL) or in a final volume of 100 µL for 24 h.

Measurements of intracellular ROS levels in NCTC2544 and HuDe cells were made using 2′, 7′-dichlorodihydrofluorescein diacetates (DCFH2-DA). DCFH2-DA and its aceto-methyl ester H2DCFDA-AM can diffuse through the cell membrane and become enzymatically hydrolyzed by intracellular esterase to produce nonfluorescent DCFH2 (2′, 7′-dichlorodihydrofluorescein), which is ionic in nature and therefore trapped inside the cell. The oxidation of DCFH2 by intracellular ROS, mainly H_2_O_2_, HO•, ROO•, NO•, and ONOO− results in fluorescent 2′, 7′-dichlorofluorescein (DCF), which stains cells [52]. Measuring the fluorescence emitted from the DCF at 535 nm, after being excited at 485 nm, allowed determination of the amount of ROS formed after the oxidation process.

Cell samples were incubated in the presence of 10 µM DCFH2-DA at 37 °C in the dark for 30 min. After this time, they were centrifuged at 1200 rpm to remove the extracellular DCFH2-DA and then suspended in a new medium. The cells were exposed to 200 U/mL of IFNγ and 10^−^^4^ M of histamine in the presence or absence of bud poplar extract (5 and 50 μg/mL) for 24 h and lysed in 1 N NaOH on the following day.

### 4.15. DPPH Assay

The antioxidant activity of chitosan or CBPR films was evaluated by using the 2,2-diphenyl-1-picrylhydrazyl (DPPH) free radical scavenging assay as previously described [13]. When the DPPH reacts with an antioxidant compound, which can donate hydrogen, it is reduced. The color changes from intense violet to light yellow. The absorbance was read at 517 nm using a UV-VIS spectrophotometer (Infinite M100 Tecan). Films, hydrated with 100 μL of water, were incubated with 100 μL of the DPPH radical (25 μg/mL) in ethanolic solution. Absorbance was measured after 30 min. The negative control was prepared with water and DPPH in the absence of the disk. The following formula was used to determine the antioxidant activity:A% = 100 − [(Abs sample-Abs blank) × 100]/Abs control

### 4.16. In Vitro Fibroblast Migration Assay

Micro-Insert 4-well dishes (Ibidi) were used for the regenerative activity assay. The HuDe cells were counted and seeded at a concentration of 5 × 10^5^/mL (15 μL in each well). The supports with the cells were incubated at 37 °C for 2 h to allow the cells to adhere to the plate. Then, 2.5 mL of cRPMI were added, and cells were incubated for 48 h to reach confluence. After the formation of the monolayer, the silicone insert was removed with sterile tweezers. In the following days, CBPR films were added for 24 h at 37 °C. Films were removed, and the cell growth was monitored under the microscope (20× magnification) for 7 days until the space between the frames was closed. The distances between the frames were recorded, and photos were taken every day.

### 4.17. Statistical Analysis

Differences between CBPR-treated biofilm and untreated biofilm were compared using the Student’s *t*-test (two-tailed). * *p*-values of <0.05 were considered significant.

## 5. Conclusions

The antioxidant and antimicrobial effects of chitosan-based films embedded with bud poplar resins were investigated. With the increase in the prevalence of wounds in almost every country, there is a growing interest in the search for natural healing agents with low cost and toxicity for the treatment of wounds and ulcers. There is indeed a great demand for wound dressing. Considering the overall results, skin application of chitosan films loaded with BPE seems to be a promising vegan approach to prevent and control wounds and acute and chronic infections.

## Figures and Tables

**Figure 1 molecules-27-07757-f001:**
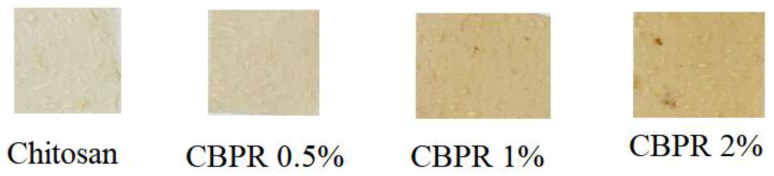
Photographs of the prepared films.

**Figure 2 molecules-27-07757-f002:**
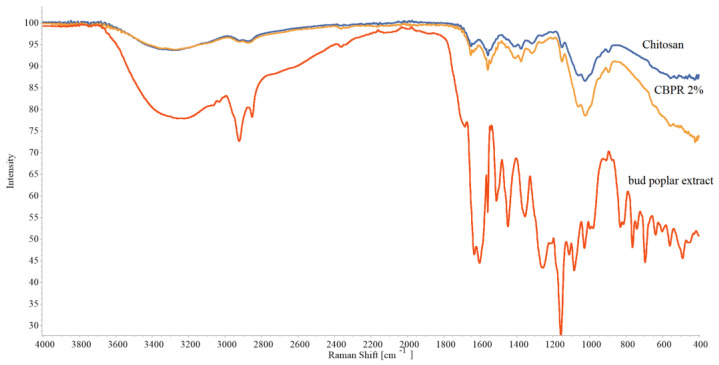
ATR-FT IR spectra of chitosan film, CBPR 2%, and bud poplar extract.

**Figure 3 molecules-27-07757-f003:**
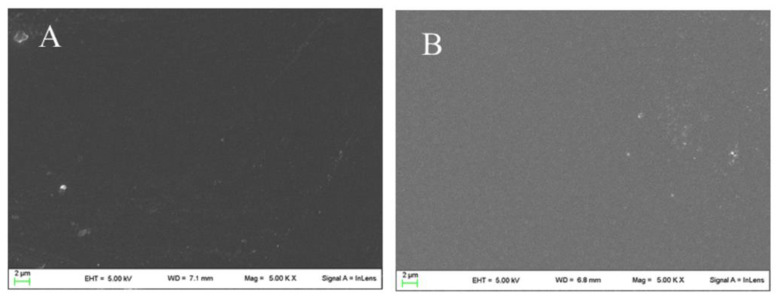
SEM micrographs of Chitosan (**A**) and CBPR 2% (**B**).

**Figure 4 molecules-27-07757-f004:**
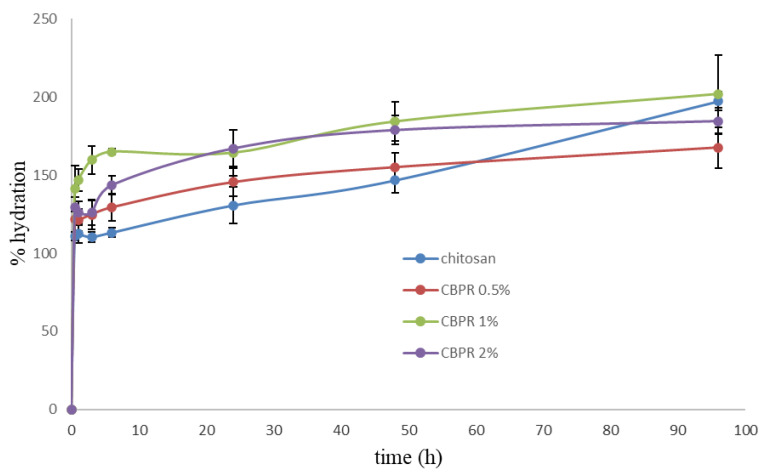
Percentage of hydration of the CBPR films as a function of time. Results are reported as an average ± SD.

**Figure 5 molecules-27-07757-f005:**
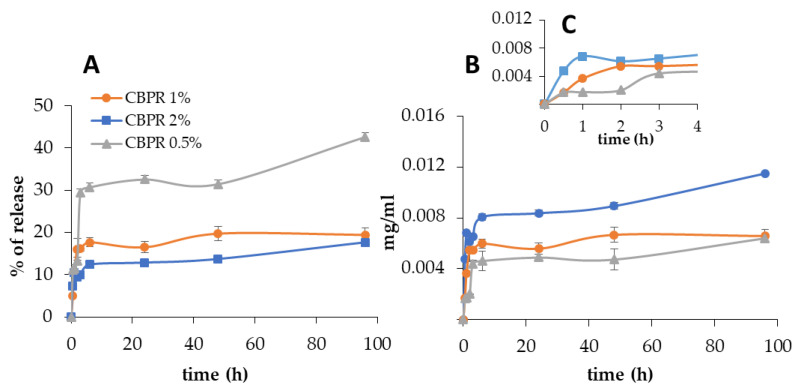
In vitro bud poplar resin release from CBPR films. Data are expressed as percentage of release (**A**) and as relative mg/mL of bud poplar resin extract in the release medium (**B**) with inset first 4 h of test (**C**).

**Figure 6 molecules-27-07757-f006:**
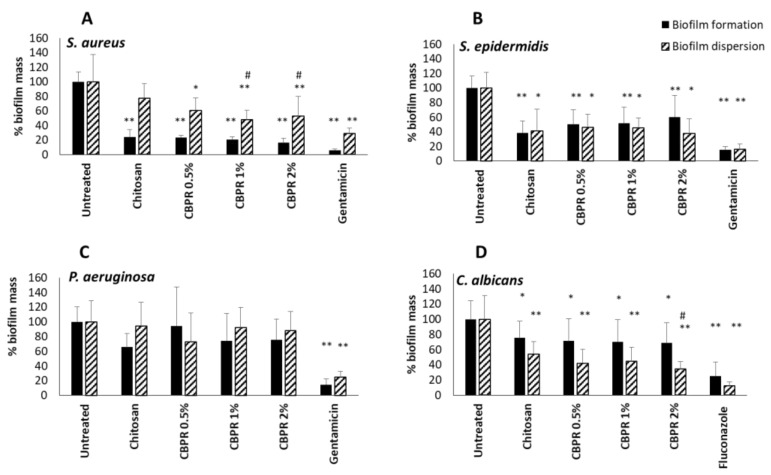
Effect of chitosan and CBPR films on the formation and dispersion of *S. aureus* (**A**), *S. epidermidis* (**B**), *P. aeruginosa* (**C**), and *C. albicans* biofilm (**D**). The data are expressed as a mean ± SD of the percentage of the biofilm mass (n = 12). The statistical analysis was performed with two-tailed Student’s *t*-test. * *p* ≤ 0.05, ** *p* ≤ 0.01 (chitosan- and CBPR-film-treated cells vs. untreated cells), # *p* ≤ 0.05 (CBPR-film-treated cells vs. chitosan-treated cells).

**Figure 7 molecules-27-07757-f007:**
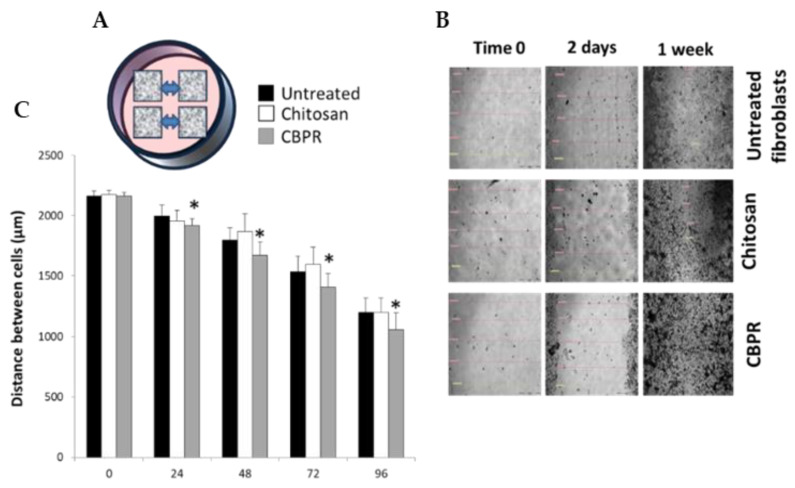
Regenerative effect of chitosan films on human dermal fibroblasts. The regenerative activity assay was performed in Micro-Insert 4-well dishes (**A**), the distances between the frames was monitored under the microscope (20× magnification) for 7 days and photos were taken (**B**). The distances between the frames were recorded and the results are expressed as mean ± SD of the measurements by microscope (n = 6) carried out in two individual experiments (**C**). the statistical analysis was performed with a two-tailed Student’s *t*-test. * *p* ≤ 0.05 (chitosan- and CBPRfilm-treated cells vs. untreated cells).

**Figure 8 molecules-27-07757-f008:**
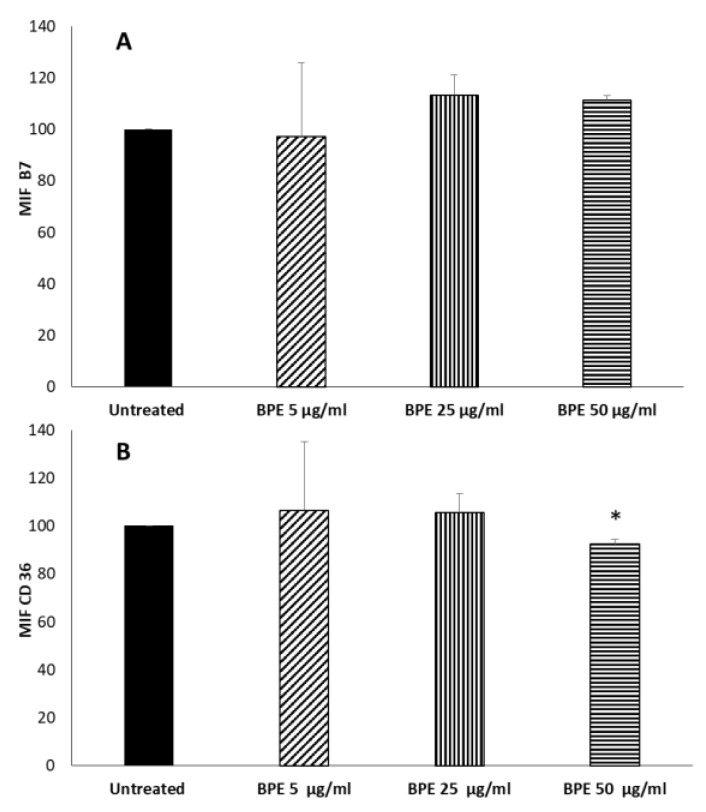
Effect of BPE on macrophages polarization. THP-1-derived macrophages were treated for 24 h with the indicated concentration of BPE. After incubation, cells were labelled with FITC mouse antihuman CD36 (**A**) and PE mouse antihuman B7 (**B**). Labelled cells were analyzed by cytofluorimeter. Results are expressed as Mean Intensity Fluorescence (MIF). The statistical analysis was performed with a two-tailed Student’s *t*-test. * *p* < 0.05 (BPE-treated cells vs. untreated cells).

**Figure 9 molecules-27-07757-f009:**
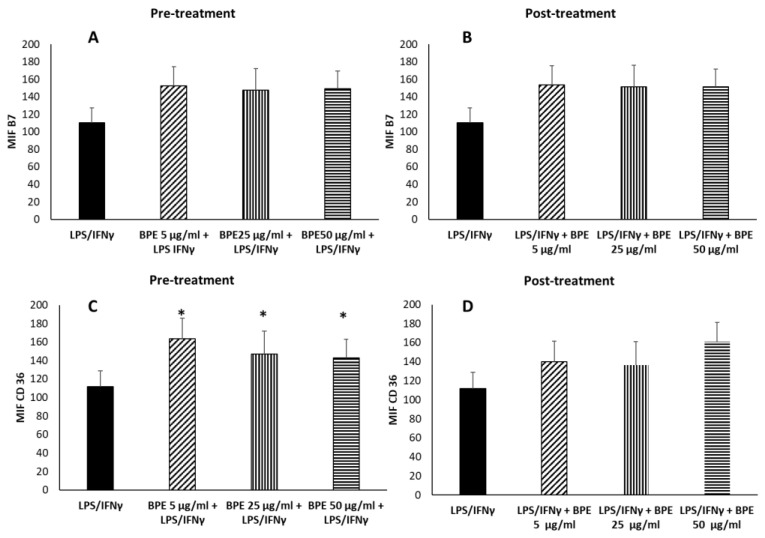
Effect of BPE pretreatment (**A**,**C**) and posttreatment (**B**,**D**) on macrophages’ polarization induced by LPS/IFNγ. THP-1-derived macrophages were pretreated for 24 h with the indicated concentrations of BPE before LPS/IFNγ stimulation (**A**,**C**) or posttreated with BPE after LPS/IFNγ (**B**,**D**). After stimulation, cells were labelled with FITC mouse antihuman B7 and PE mouse antihuman CD36. Labelled cells were analyzed by cytofluorimeter. Results are expressed as Mean Intensity Fluorescence (MIF). The statistical analysis was performed with a two-tailed Student’s *t*-test. * *p* < 0.05 (BPE+LPS/IFNγ-treated cells vs. LPS/IFNγ-treated cells).

**Figure 10 molecules-27-07757-f010:**
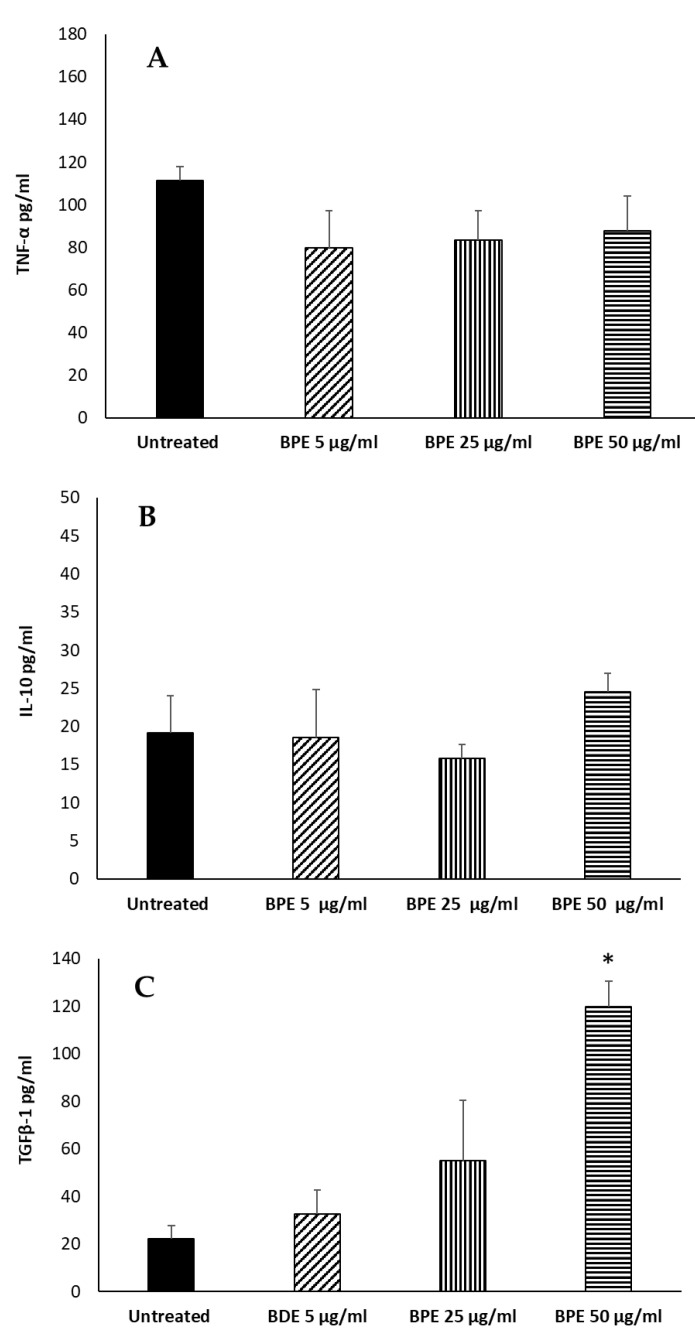
Effect of BPE on TNF-α (**A**), IL-10 (**B**), TGFβ1 (**C**) production by macrophages. THP-1-derived macrophages were stimulated with BPE. Data represent the mean ± standard deviation (SD) of three independent experiments. The statistical analysis was performed with a two-tailed Student’s *t*-test. * *p* < 0.05, (BPE-treated cells vs. untreated cells).

**Figure 11 molecules-27-07757-f011:**
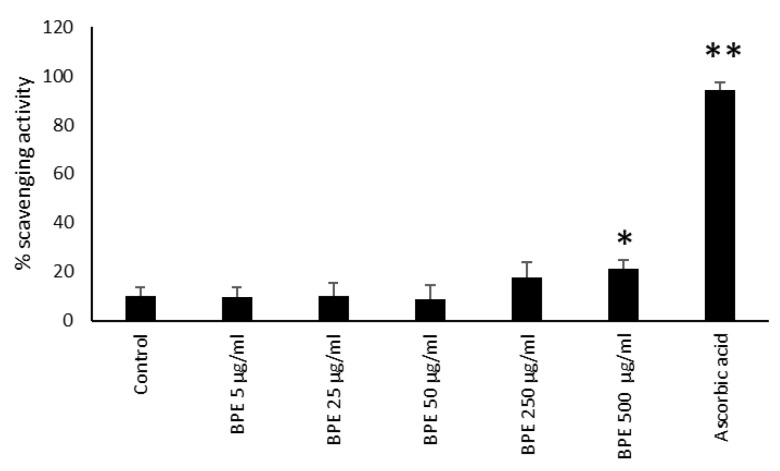
BPE radical scavenging capacity by DPPH method. Results are expressed as percentage of DPPH scavenging activity. Data represent the mean ± standard deviation (SD) of two independent experiments performed in triplicate. The statistical analysis was performed with a two-tailed Student’s *t*-test. * *p* < 0.05, ** *p* < 0.01 (BPE-, ascorbic-acid- treated cells vs. ethanol-treated cells).

**Figure 12 molecules-27-07757-f012:**
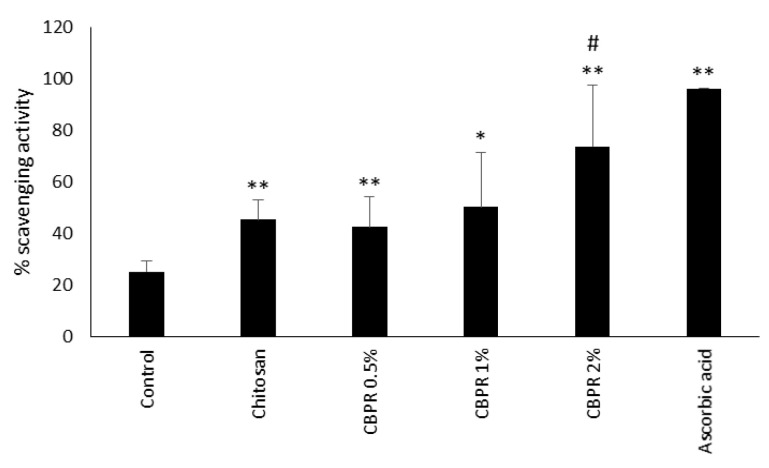
Scavenging activity of chitosan and CBPR. The data are expressed as a mean ± SD of the percentage of the scavenging activity (n = 6). The statistical analysis was performed with a two-tailed Student’s *t*-test. * *p* ≤ 0.05, ** *p* ≤ 0.01 (film-treated sample vs. control), # *p* ≤ 0.05 (chitosan/CBPR-film treated samples vs. chitosan-treated samples).

**Figure 13 molecules-27-07757-f013:**
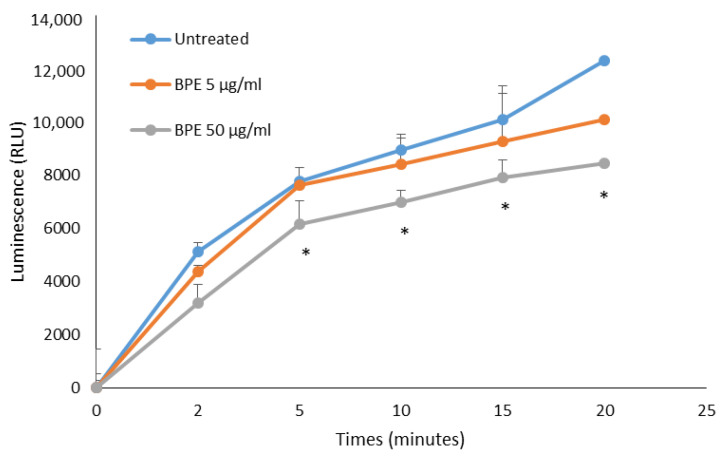
In vitro neutralization of ROS produced by human PMA-stimulated PMN. Results are expressed in relative luminescence units (RLU). Data represent the mean of three independent experiment performed in triplicate for BPE at two different concentrations. The statistical analysis was performed with a two-tailed Student’s *t*-test. * *p* < 0.05 (BPE-treated cells vs. untreated cells).

**Figure 14 molecules-27-07757-f014:**
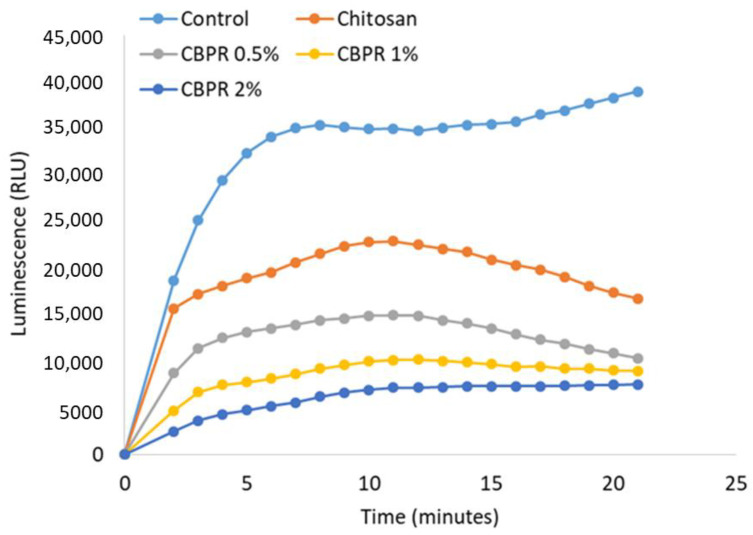
Antioxidant activity of different chitosan/BPE films on ROS production in human neutrophils. Results are expressed as RLU (relative luminescence units). The results are expressed as mean ± SD of the measurements made under the microscope (n = 6) carried out in two individual experiments.

**Figure 15 molecules-27-07757-f015:**
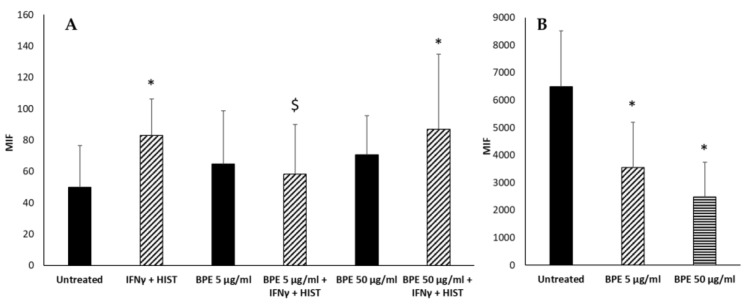
Effect of BPE on intracellular ROS production of HuDe (**A**) and NCTC 2544 (**B**) cells. The experiment was performed in quadruplicate for each substance. The statistical analysis was performed with a two-tailed Student’s *t*-test. * *p* < 0.05 (BPE-treated cells vs. untreated cells); $ *p* < 0.05 (BPE-treated cells vs. IFNγ+-HIST-treated cells).

**Table 1 molecules-27-07757-t001:** Antimicrobial activity evaluated by Kirby–Bauer assay. The diameter of the inhibition zone reported is the mean of three measures of three independent experiments.

Treatments	Inhibition Halo Diameter (mm)			
	*Staphylococcus aureus*	*Staphylococcus epidermidis*	*Pseudomonas aeruginosa*	*Candida albicans*
**Chitosan**	7.5	7.5	6	7.5
**CBPR 0.5%**	7	7.5	7	7.5
**CBPR 1%**	8	10	7.5	7.5
**CBPR 2%**	8.5	11.5	8	8
**Gentamicin/Fluconazole**	22	28	24	8

**Table 2 molecules-27-07757-t002:** Cytotoxicity of CBPR films on human fibroblasts and keratinocytes. Results are expressed as percentage of live cells with respect to untreated cells, assumed to be 100. The results are expressed as mean ± SD of four different measures.

	Hude		NCTC2544	
	4 h	24 h	4 h	24 h
**Untreated**	100 ± 12	100 ± 9	100 ± 25	100 ± 39
**Chitosan**	85 ± 30	85 ± 4	68 ± 30	65 ± 17
**CBPR 0.5%**	73 ± 23	67 ± 26	125 ± 33	57 ± 14
**CBPR 1%**	84 ± 25	63 ± 4 **	53 ± 28 *	27 ± 19 *
**CBPR 2%**	67 ± 40	45 ± 26 *	36 ± 8 **	29 ± 22 *

The statistical analysis was performed with a two-tailed Student’s *t*-test. * *p* ≤ 0.05, ** *p* ≤ 0.01 (film treated cells vs. untreated cells).

## Data Availability

All data used to support the findings of this study are available from the corresponding author upon request.

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
