# Peer review of "Bud-Poplar-Extract-Embedded Chitosan Films as Multifunctional Wound Healing Dressing"

_molecules, 2022, doi:10.3390/molecules27227757_

Round 1

Reviewer 1 Report

The work presents chitosan films embedded with bud poplar extract (BPE) as wound healing dressing. The authors present interesting results regarding the antimicrobial activity of the films and their effect on fibroblast migration, all from in vitro assays. The paper is well written, but presents some inconsistencies that the authors should correct or better explain before the article can be published:

1.           Table 2 shows a high cytotoxicity of the chitosan / BPE composites at 24 hours, if the test were performed at a somewhat longer time the result would probably lead to think that the materials are not suitable for the indicated application.  

2.           Actually in the work two types of results are presented: a part of the tests are performed on chitosan films containing 0.5, 1 or 2% (w/w) of BPE (antimicrobial activity, fibroblast migration, antioxidant activity), while others analyze the effect of BPE alone, which would simulate the effect of BPE released from the film (anti-inflammatory activity, antioxidant activity). However, in the case of BPE in suspension, concentrations up to 0.05 mg/mL are used, (even in Figure 8 the effect of concentrations up to 0.5 mg/mL is shown)  while Figure 2b shows that BPE release from the films reaches concentrations of only 0.01 mg/mL and that in the case of the maximum concentration of BPE in the chitosan film which, as seen in Table 2, is highly cytotoxic and in a medium containing ethanol, which facilitates the release. The authors should discuss in detail the choice of concentrations used and ensure that they are meaningful for the situation that would be encountered in vivo.

3.           Water uptake and BPE release characterization tests have been performed up to 100h which is insufficient time in both cases for an equilibrium situation to be reached. The trend shown in the figures is indicative of a high uncertainty in the measurements and a statistical analysis has not been carried out to determine whether the differences found between one sample and another are statistically significant or whether the trend marked by the lines drawn in the figures is significant.

4.           The conclusion expressed in lines 623 and following cannot be expressed independently of the concentration of BPE introduced in the chitosan film and its cytotoxicity.

Author Response

Reviewer 1

The work presents chitosan films embedded with bud poplar extract (BPE) as wound healing dressing. The authors present interesting results regarding the antimicrobial activity of the films and their effect on fibroblast migration, all from in vitro assays. The paper is well written, but presents some inconsistencies that the authors should correct or better explain before the article can be published:

  1. Table 2 shows a high cytotoxicity of the chitosan/BPE composites at 24 hours, if the test were performed at a somewhat longer time the result would probably lead to think that the materials are not suitable for the indicated application.  

The aim of determining the cytotoxicity of CBPR films on fibroblasts and keratinocytes was to select the safe CBPR film to be tested in the regeneration test in which the contact of cells with film was of 24h. The results obtained led to the choice of the CBPR 0.5 film. The evaluation of the regenerative activity was performed only on the fibroblasts since the keratinocytes under our experimental conditions underwent a progressive reduction of vitality.

  1. Actually in the work two types of results are presented: a part of the tests are performed on chitosan films containing 0.5, 1 or 2% (w/w) of BPE (antimicrobial activity, fibroblast migration, antioxidant activity), while others analyze the effect of BPE alone, which would simulate the effect of BPE released from the film (anti-inflammatory activity, antioxidant activity). However, in the case of BPE in suspension, concentrations up to 0.05 mg/mL are used, (even in Figure 8 the effect of concentrations up to 0.5 mg/mL is shown)  while Figure 2b shows that BPE release from the films reaches concentrations of only 0.01 mg/mL and that in the case of the maximum concentration of BPE in the chitosan film which, as seen in Table 2, is highly cytotoxic and in a medium containing ethanol, which facilitates the release. The authors should discuss in detail the choice of concentrations used and ensure that they are meaningful for the situation that would be encountered in vivo.

In figure 8 (the current figure 9) we studied the anti-oxidant activity of BPE extract in free-cell assay; we test increasing concentrations of BPE until we could relieve an effect. In Fig 9 (the current figure 10), we tested CBPR films where we observed an antioxidant activity of chitosan itself and the CBPR 2% showed a significant increase of antioxidant activity with respect to chitosan.

The cytotoxicity of CBPR films showed a cytotoxic effect at 4 and 24h due to chitosan film. The observed effect is probably attributable to the direct contact of film with the cell monolayer (85% and 65% of life Hude and NCTC2544 cells, respectively). Addition of BPE extract to chitosan increase the cytotoxicity but it is difficult to ascribe the cell vitality reduction to BPE extract, chitosan or to their combination. Furthermore, in a previous work (De Marco S, 2017) the cytotoxic activity of bud poplar extract on human PBMCs has been evaluated, obtaining a 24-hour CC50 of 88.5 µg/ml. Based on this data the effect of the extract on anti-inflammatory activity was tested at concentrations of 5-25 and 50 µg/ml. The cytotoxicity on THP-1-derived macrophages untreated or treated with BPE was determined and BPE was not cytotoxic. This is the reason why we did not add the figure in the first manuscript. The results were added as supplementary data (Figures 2S) in the result section in 2.4 Anti-inflammatory activity of BPE) paragraph.

  1. Water uptake and BPE release characterization tests have been performed up to 100h which is insufficient time in both cases for an equilibrium situation to be reached. The trend shown in the figures is indicative of a high uncertainty in the measurements and a statistical analysis has not been carried out to determine whether the differences found between one sample and another are statistically significant or whether the trend marked by the lines drawn in the figures is significant.

As suggested by the Referee, figures 1 and 2 were revised and SD values have been added. As concerns the time of the experiment, Figure 1 (current Figure 2) shows that after 100 h the hydratation percentage reaches a plateau in the case of Chitosan/propolis extract films, thus it was not necessary to prolong the test lenght. As concerns the release test, the lenght of the test is higher than that of biological tests and thus it is enough to assert that the biological and antimicrobial results are due to the released bud popular resin.

  1. The conclusion expressed in lines 623 and following cannot be expressed independently of the concentration of BPE introduced in the chitosan film and its cytotoxicity.

We agree with the reviewer on this point. The conclusions have been revised as suggested.

Reviewer 2 Report

Dear  Authors,

This is a paper devoted to studying the Bud poplar extract embedded chitosan films as a multifunctional wound healing dressing. Several issues must be attended to before the manuscript would be considered for publication.

Abstract.

Must be improved including more quantitative results and no only descriptive ones.

In line 30...vegan word is not a proper concept for describing this biocomposite...it is a food concept.... it can be replaced with... plant-based...or another phrase.

Introduction.

Large is ok, it is informative but maybe for comparison, propolis is not addecuate name because it is produced only after bee zucking and processing the flowers nectar. Maybe it would be better if it would be compare with other natural extracts, including the one's from other parts from the same plants.  

Results

The main flaw of this manuscript is the lack of chemical characterization of films to demonstrate they chemical identity (FTIR, SEM, TGA, XRD), mechanical, transparency, etc)

In figure 1. A small graph showing the kinetics of water absorption of the first part of the curve (0-30min) can be inserted into this graph.

Be consistent in the whole manuscript. As the description of the bud poplar extract is not presented, you must say it is a plant extract, why do you describe as propolis...it was isolated from honey bee ?....

Line 145 to 150 are methodology, just move some parts of them to this manuscript section.

 In Table 1. First column label is wrong,,,,it belongs to Products or Treatments. The former label can be placed over the names of the bacterial strains.

Figure 3. Each graph must be labeled with different letters and described in the figure caption. 

Figure 4. There are pictures with different letters but in graph, captions were not identified.

Figure 5. the same than Figure 3, even more, the option must be placed just below the graphics.

In all graphics or tables indicates which statistical test were used. Also, during the results description , you must mention (in case of corresponding) that there are significant differences between treatments. 

A discussion of all topics must be done, including references and comparing with those previously published from both all and fresh literature.

Conclusiong. Must be addecuate to the 

Author Response

Reviewer 2

Dear  Authors,

This is a paper devoted to studying the Bud poplar extract embedded chitosan films as a multifunctional wound healing dressing. Several issues must be attended to before the manuscript would be considered for publication.

Abstract.

Must be improved including more quantitative results and no only descriptive ones.

We thanks the reviewer for its suggestion The abstract has been modified adding the results.

In line 30...vegan word is not a proper concept for describing this biocomposite...it is a food concept.... it can be replaced with... plant-based...or another phrase.

The “word vegan” has been replaced with “plant–based” as suggested.

Introduction.

Large is ok, it is informative but maybe for comparison, propolis is not addecuate name because it is produced only after bee zucking and processing the flowers nectar. Maybe it would be better if it would be compare with other natural extracts, including the one's from other parts from the same plants.  

As reported in previous papers (De Marco S, 2017; Salatino A, 2022), propolis and poplar bud exudates have the same ingredients even if the composition can be different. In fact, it has been demonstrated that European propolis contains flavonones, flavones, phenolic acids, and in the Brazilian propolis are diterpenes, and flavonoids are present. Similarly, the poplar buds are coated with a resin that contains different phenolic compounds as terpenoids, flavonoid aglycones, and their chalcones and phenolic acids and their esters.

Results

The main flaw of this manuscript is the lack of chemical characterization of films to demonstrate they chemical identity (FTIR, SEM, TGA, XRD), mechanical, transparency, etc)

In this paper, our attention was pointed to biological and antimicrobial properties of films. As far as it concerns film characterization, in the paragraph 2.1 the film thickness and film photographs (Figure 1) have been introduced. Moreover, ATR- FTIR spectra of films were performed and they are reported in Supplementary information. We agree with the reviewer with the other characterizations of the films suggested but it was not possible to carry them out in the time for the review. In addition, we should have involved other colleagues who have the facilities.

In figure 1. A small graph showing the kinetics of water absorption of the first part of the curve (0-30min) can be inserted into this graph.

The hydration of films was detected starting from 30 minutes of immersion as in most papers, which describe hydration properties of wound films. A small graph showing the kinetics of water absorption of first 4 hours has been added in the figure 1 (Current Figure 2).

Be consistent in the whole manuscript. As the description of the bud poplar extract is not presented, you must say it is a plant extract, why do you describe as propolis...it was isolated from honey bee ?....

Given the similarity of the composition of propolis and poplar bud extracts, our research project aimed to investigate whether the poplar bud extract had biological activities comparable to those of propolis. Many researches done in the past refers to propolis and a few researches to poplar buds. In discussing our data, we compared our results obtained with BPE with those obtained by other authors with propolis and poplar buds.

Line 145 to 150 are methodology, just move some parts of them to this manuscript section.

The Kirby Bauer method has been moved to Materials and Methods, Kirby Bauer assay section (4.5).

 In Table 1. First column label is wrong,,,,it belongs to Products or Treatments. The former label can be placed over the names of the bacterial strains.

The table 1 has been modified as suggested.

Figure 3. Each graph must be labeled with different letters and described in the figure caption. 

Each graph of Figure 3 (current Figure 4) have been labelled and described in the Figure caption.

Figure 4. There are pictures with different letters but in graph, captions were not identified.        

The Figure 4 (current figure 5) has been described in the caption.

Figure 5. the same than Figure 3, even more, the option must be placed just below the graphics.

Each graph of Figure 5 (current Figure 6) have been labelled and indicated in the Figure caption.

In all graphics or tables indicates which statistical test were used. Also, during the results description, you must mention (in case of corresponding) that there are significant differences between treatments.

The statistical test used has been added to the caption of all the figures.

A discussion of all topics must be done, including references and comparing with those previously published from both all and fresh literature.

The antimicrobial activity was discussed in the second paragraph of discussion section. As explained in the third paragraph, to the best of our knowledge the antibiofilm and cell proliferative activities of chitosan BPE films have not been evaluated. We reported the anti-inflammatory activity of propolis described by Bueno-Silva (2015) and Conti (2015); the results of anti-oxidant activity of CBPR have been compared with the propolis and poplar bud extract activities described by Moreno (2000), Kurek-Gorecka (2012), Bonamigo (2017), Dudonne (2011) and Poblocka-Olech (2019).

Conclusiong. Must be addecuate to the 

We think that the reviewer’s comment is not complete.

Round 2

Reviewer 2 Report

Dear authors,

I suggest the FTIR spectra of samples must be placed in the main test (it is the only chemical characterization you provided). Also, the discussion of this part must be done properly. Now, it is very general, which bands appear or disappear during composite preparation?. Which bands are characteristics of chitosan and plant-based extracts?. 

For wound healing dressing you choose chitosan because it is a good matrix, but more characterization technique is necessary. The biological activity is well performed but the physicochemical characterization of the matrix is poor.... maybe SEM or TGA can allows you improve this part. 

Author Response

I suggest the FTIR spectra of samples must be placed in the main test (it is the only chemical characterization you provided). Also, the discussion of this part must be done properly. Now, it is very general, which bands appear or disappear during composite preparation?. Which bands are characteristics of chitosan and plant-based extracts?. 

As suggested, FTIR spectra of samples has been reported in the main text (New Figure 2) and a discussion has been introduced.

For wound healing dressing you choose chitosan because it is a good matrix, but more characterization technique is necessary. The biological activity is well performed but the physicochemical characterization of the matrix is poor.... maybe SEM or TGA can allows you improve this part. 

In order to improve physicochemical characterization, SEM micrographs have been reported in New Figure 3 in the main text.